# Dimensionally Stable Delignified Bamboo Matrix Phase-Change Composite under Ambient Temperature for Indoor Thermal Regulation

**DOI:** 10.3390/polym15071727

**Published:** 2023-03-30

**Authors:** Qinchen Duan, Xin Zhang, Shuang Lang, Guowei Liu, Hui Wang, Xiaojian Zhou, Guanben Du

**Affiliations:** Key Laboratory of State Forestry and Grassland Administration on Highly-Efficient Utilization of Forestry Biomass Resources in Southwest China, Yunnan Key Laboratory of Wood Adhesives and Glued Products, Southwest Forestry University, Kunming 650224, China

**Keywords:** delignified bamboo, polyethylene glycol, phase-change temperature, energy storage, dimensional stability

## Abstract

Energy storage materials to modulate indoor microclimates are needed to improve energy efficiency and for human comfort. Of these, phase-change material (PCM) is considered a very useful material because of its excellent latent heat energy storage. For application, some synthetic porous materials for supporting PCM are usually not friendly enough for people and housing environments due to their non-degradation characteristics. Hence, to develop an eco-friendly porous material is needed in order to encapsulate PCM composites that are always expected in indoor applications. In this work, heat-treated bamboo bricks were delignified to provide a delignified bamboo (DB) matrix. A phase-change composite was then fabricated by impregnating DB with polyethylene glycol (PEG) polymer. Impregnation was carried out under wet conditions to ensure the regular arrangement of the DB structure so as to achieve dimensional stability. The final DB/PEG composite was investigated for dimensional stability, load rate, latent heat, and phase-change temperature. Results showed that the DB matrix could be easily impregnated with PEG polymer under wet conditions, and the DB/PEG composite was found to have high enthalpy and a large phase-change temperature interval. Moreover, the composite was found to be a good regulator of indoor temperature and a stable dimension with a snow-white appearance. In summary, this DB/PEG composite is an energy storage material with the potential to modulate ambient indoor temperature and reduce building energy consumption.

## 1. Introduction

Energy shortages and environmental pollution have become serious problems for the development of human society, with continuous consumption of fossil resources. Meanwhile, a large amount of recyclable thermal energy is wasted. Developing renewable sources and improving energy efficiency are considered the main means to solve this problem. In recent year, energy storage technologies based on phase-change materials (PCMs) have attracted extensive research interest: PCMs could store or convert thermal energy from surrounding environmental irradiation and waste heat by going through a phase-change process at a relatively constant temperature [1,2]. With the advantages of high energy density, adjusting ambient temperature, and reducing energy consumption, PCMs have been widely used in green building construction, cylindrical power battery packs, storage boxes for food, intelligent temperature control clothing, and so forth [3].

In general, PCMs can be categorized into organic, inorganic, and organic–inorganic mixtures; among them, organic PCMs are the most promising owing to their high latent heat energy storage [4,5,6]. Unfortunately, leakage issues with organic PCMs restrict their application during the phase transition process. To address this problem, the introduction of a supporting porous material, such as metal foam, polymeric materials, aerogels, porous cellulose films, etc. [7,8,9], can stabilize the PCM matrix. Compared with synthetic porous materials, natural porous polymers such as wood and bamboo are gradually gaining more attention owing to their being renewable, recyclable, abundant, and environmentally friendly.

PCMs impregnated using delignified wood have a unique porous structure that prevents against leakage, and can provide a comfortable indoor environment. Bamboo has a similar structure and chemical composition to wood [10,11,12,13,14], but has a higher growth rate and yield measured per acre. Therefore, bamboo is thought to be a promising material to replace wood in many fields. From a chemical perspective, bamboo is composed of cellulose, hemicellulose, and lignin, among which lignin could be utilized as a binder and dispersant. After removing lignin, bamboo presents higher porosity, better defined channels, and a larger specific surface area compared with natural bamboo, and is an excellent candidate to stabilize PCMs. For obtaining perfect porous structure from bamboo, a variety of delignified methods have been exploited [15,16,17], among which the chemical treatment method is still the most effective for bamboo treatment due to its unique structure. Although the treatment with chemical reagent is not environmentally friendly, it is one step during the delignification process, and with subsequent steps proceeding, the final products are not toxic. Hence, chemical treatment is used extensively, especially for treating large size bamboo to accommodate widespread applications for bamboo.

Nowadays, a number of researchers are working on phase-change material composites by delignified wood as the supporting porous material. The primary applications of wood material in phase-change materials design and fabrication are those based on wood that has been delignified. The first reported delignification wood-based phase-change composites were created by simply combining delignification wood with phase-change materials, to fulfill various temperature ranges for applicability in various fields. Various functional modification based on the wood-based PCMs derived from delignified wood frame was achieved. At the same time, these enhancements were created based on fulfilling the phase-change function. For example, the PCMs with self-cleaning function was created by spraying modified silica in an epoxy resin/acetone solution [13]; the self-luminous function was created by adding carbon quantum dots [10]; and the thermochromic function was created with the phase-change material composed of 1-Tetradecanol, double A, and crystal ultraviolet lactone [14]. As a base stage, the delignified bamboo served as the framework to prepare phase-change materials was frequently overlooked despite being the most important. Functional experiments on phase-change materials based on delignified bamboo have only ever detected heat discoloration [18]. Therefore, it is essential to do the research which was described simple delignification bamboo-based phase-change materials, because there are not many studies on these kinds of materials by using delignification bamboo framework thereafter been functionalized. The fundamental research on the phase-change materials currently based on delignification bamboo should to do a depth exploration, in order to move on to in-depth functional research.

As a typical PCM, polyethylene glycol (PEG) is a commonly used material that is cost-effective, has good thermal stability and high latent enthalpy, and is non-toxic and non-corrosive [19]. Moreover, PEG has a broad phase-transition temperature range controlled by changing the molecular weight [20]. As such, the low molecular weight PEG polymer often shows good potential as a material for creating comfortable indoor environments. Moreover, PEG possesses the characteristics of polyhydroxy, which has a significant attraction for delignified bamboo using delignification technology. Thereby, in order to more easily combine the two materials, the bamboo goes through a delignification procedure that enables it to shed the majority of its lignin and expose the polyhydroxy of cellulose. Finally, Due to straightforward combining capabilities and the ability to control temperature through variations in molecular weight, the PEG is employed.

In this work, heat-treated bamboo (HTB) bricks were selected for removing lignin much more easily compared with original bamboo to obtain an excellent delignified bamboo (DB) matrix; then, a composite was fabricated by impregnating the DB with PEG (Mn 800/Mn 1000) polymer for obtaining dimensionally stable phase-change composites. From the results obtained in this work, the prepared phase-change composite material can reduce energy consumption by storing thermal energy, and also could potentially be applied as a indoor energy storage material.

## 2. Materials and Methods

### 2.1. Materials

Heat-treated moso bamboo (*Phyllostachys edulis*) bricks were purchased from a factory in Zhejiang Province, China; the moisture content was 8.7%. PEG (Mn = 800, >99%) and PEG (Mn = 1000, >99%) were obtained from the Guangfu Fine Chemical Research Institute in Tianjin, China. Chemical reagents, including sodium hydroxide (NaOH, >96%), glacial acetic acid (CH_3_COOH or HAC >99.5%), hydrogen peroxide solution (H_2_O_2_, 30%), sodium chlorite (NaClO_2_, 80%), and sodium hypochlorite (NaClO, 10%), were supplied by the Aladdin Biochemical Technology Co., Shanghai, China.

### 2.2. Delignification and Bleaching of Bamboo Samples

First, heat-treated bamboo (HTB) was selected and cleaved into chips (50.0 mm long, 21.0 mm wide, and 5.3 mm thick), and placed into boiling sodium hydroxide (5 wt.%) for 8 h to remove lignin. Then, these delignified bamboo samples were washed in distilled water, and bleached with a solution containing HAC (1 wt.%), sodium chlorite (5 wt.%), and sodium hypochlorite (3.5 wt.%) for 8 h. Afterwards, the samples were further immersed in hydrogen peroxide (1 wt.%) solution at 85 °C for 4–5 h to eliminate the remainder of the mixture solution. Finally, this bamboo cellulose with an integral bio-matrix structure was washed at least three times with distilled water.

### 2.3. Preparation of Bamboo Cellulose Matrix Phase-Change Composite

The bamboo cellulose matrix phase-change composite was prepared as follows (Figure 1): PEG-800 was melted at 60 °C, and the delignified bamboo (DB) was immediately dipped into the PEG-800 melt liquid. It was then evacuated under 200 Pa for 60 min at 60 °C until the DB was filled with PEG-800. After vacuum treatment was completed, excess PEG-800 adhering to the DB surface was removed by washing with ethanol absolute, then drying for 24 h at room temperature. The resulting bamboo cellulose matrix phase-change composite was named DB/PEG-800. To adapt to the indoor environment and avoid PEG leakage during the phase-change process, the final DB/PEG phase-change composite was fabricated using DB/PEG-800 immersed again in melted PEG-1000 liquid, then treated under vacuum conditions under 200 Pa at 80 °C for 60 min with the same process as that of DB/PEG-800. The apparatus used for vacuum treatment was the Fujiwara vacuum pump.

### 2.4. Characterization

#### 2.4.1. Lignin Content Testing

The lignin content of HTB and DB was tested according to the standard for Klason lignin (TAAPI T222 om-02). For this, 1.0 g HTB and DB powder were filtered using a mixture of ethanol and benzene for 6 h at 85 °C, and then dried naturally for 24 h. The obtained specimen quality was named m0. Next, the material was treated with 15 mL of 72% sulfuric acid for 2 h at 20 °C; the mixture was diluted to 3% concentration of sulfuric acid by adding 560 mL distilled water and boiling for 4 h. After cooling to room temperature, the mixture was filtered and washed with distilled water and the weight of insoluble materials was named m1. Finally, the ash weight m2 was determined by the standard of TAAPI T211 om-02: that is, m1 was burned in a furnace.

The lignin content (%) was determined as = [(m2−m1)/m0] × 100% [15].

#### 2.4.2. Cellulose and Hemicellulose Content Testing

The contents of cellulose and hemicellulose in HTB and DB samples were confirmed via the weight difference method. Firstly, 1.5 g dried HTB and DB powder was dissoluted using 150 mL 2 mol/L hydrochloric acid for 48 min at 105 °C to remove hemicellulose and then was filtered with ethanol 95%, anhydrous ethanol, and acetone twice in turn, then dried at 60 °C until its weight was constant. The obtained remaining quality was named m3 [21]. Secondly, the remaining substance was processed with 15 mL of 75% sulfuric acid for 3 h at room temperature, then it was diluted with 135 mL distilled water and stayed one night at room temperature. The materials were immersed in sulfuric acid, then filtered and washed with distilled water, and then dried at 60 °C, and the obtained substance was named m4.

The hemicellulose content (%) was determined as = [(1.5−m3)/1.5] × 100%.

The cellulose content (%) was determined as = [(1.5−m3−m4)/1.5] × 100%.

#### 2.4.3. Scanning Electron Microscopy (SEM)

The samples were cut from the lateral direction with the method of brittle fracture and sprayed with gold atom under vacuum. The morphology of the (OB), HTB, DB/PEG composite cross-section was detected via scanning electron microscopy (SEM, TESCAN MIRA LMS, Brno, Czech Republic).

First, the sample of HTB was cut with a small sawing machine and the size was 5.0 mm long, 5.0 mm wide, and 5.3 mm thick. Then the same size of DB sample was created with that of above HTB. Finally, the sample of DB was placed in water and allowed to freeze. After that, cryogenic SEM was used to obtain the DB micro-images (Cryo-SEM, FEI Quanta 450 equipped with Quorum PP3000T, Quorum Technologies, Laughton, UK).

#### 2.4.4. X-ray Diffractometer (XRD)

The samples were made into abrasive powders and placed on a slide. Then, the samples were tested via X-ray diffractometer (XRD, ULTINA IV, Rigaku Tehnologies Inc., Austin, TX, USA) between 5° and 90°. Finally, the crystal structure of the samples was analyzed.

#### 2.4.5. Fourier Transform Infrared Spectrometer (FT-IR)

The samples were ground into powders and mixed with potassium bromide (KBr), then ground again and placed in a manual tablet press for tableting. Next, the pressed transparent sample sheets were analyzed and recorded with the Fourier Transform Infrared spectrometer (FT-IR, NICOLET iS50 FT-IR) spectra with a wavelength range from 4000 cm^−1^ to 450 cm^−1^, allowing to analyze the changes of the functional groups in samples.

#### 2.4.6. Differential Scanning Calorimeter (DSC)

The phase-change temperature and enthalpy of samples were determined with a differential scanning calorimeter (DSC, TA Discovery DSC 250, TA Instruments, New Castle, UK). The gas test environment was nitrogen, and the testing temperature range of heating and cooling was between −30 °C and 100 °C. At the same time, a heat or cool rate was 10 °C/min.

#### 2.4.7. Thermogravimetric Analyzer (TGA)

To confirm the weight change of composite, 10 mg powder sample was measured with a thermogravimetric analyzer (TGA, DTG, STA 600) from 20 °C to 650 °C. Meantime, the nitrogen atmosphere served as the test condition, and the heating rate was 10 °C/min.

#### 2.4.8. Dimensions and Quality Recording

The shape stability of the sample was investigated by comparing the dimensions of HTB and its composite at room temperature. The quality of the sample was measured via electronic balance in order to determine how much PEG was loaded into the PEG/DB composites.

#### 2.4.9. Infrared Thermal Imager (IR Imager)

The duration and temperature change time of the DB/PEG composite was determined with an infrared thermal imager (IR imager, TUi 120S, iTherml Technology Co., Hangzhou, China), and the image of the phase-change temperature of sample was recorded between 0 °C and 40 °C. Finally, the change time of the samples was determined depending on their temperature.

#### 2.4.10. The Thermal Stability of DB/PEG Composite

The DB/PEG and PEG samples were put on filter paper and then placed in a drying oven to determine the thermal stability of the DB/PEG composite. The temperature was increased from 25 °C to 60 °C, the tested samples were observed and recorded at 25 °C, 40 °C, and 60 °C with photographs, respectively. Eventually, the thermal stability was assessed in accordance with the state change of the samples.

## 3. Results and Discussion

Delignified bamboo (DB) was selected as a porous supporting framework to overcome the PEG leakage problem. Compared with wood, the structure of bamboo is much denser; as it is difficult to remove lignin completely using the same method as that applied to wood, so heat-treated bamboo (HTB) was selected and a multi-step treating process was explored to obtain intact bamboo cellulose matrix. As shown in Figure 2a, before bleaching, the specimen of bamboo was brown and exhibited residual lignin. A white bamboo cellulose brick was obtained after immersion in acid mixture solution for a period of time, during which the lignin should be completely removed. Meanwhile, the DB specimen exhibited excellent light transmittance performance, offering the potential as a decorative material.

As shown in Figure 2b, the content of lignin, cellulose and hemicellulose were calculated according to their weight change. From the obtained values, after delignification, the content of lignin decreased obviously in the DB sample, and the remaining lignin content was only 3.8%, far lower than that in the HTB sample and the reported literature. This result confirmed that our delignification method was efficient for treating large volumes (>5 mm, thick) of bamboo [15]. Furthermore, the ratio of cellulose and hemicellulose in DB was increased remarkably compared with HTB, indicating that a part of the hemicellulose was removed as well during the delignification process.

FT-IR spectra (Figure 2d) show peaks of the HTB sample at 1617 cm^−1^ (C=C, stretching vibration) [16,22,23], 1508 cm^−1^ (skeleton vibration) [18], and 1401 cm^−1^ (C–H, symmetric deformed vibration) [24], based on comparison with a DB sample, suggesting that lignin and phenol extraction occurred during delignification treatment. A decline in the peak of 1737 cm^−1^ (hemicellulose of acetyl groups, C=O, stretching vibration) illustrates that hemicellulose was degraded during the delignified treatment. However, the crystalline features of DB remained prominent (Figure 2c), including two main peaks of 16° (101) and 22° (002) [25]. the peak at 35° (040) was weak. These findings confirmed that DB has a regular structure and has the potential to be an excellent framework material. 

The DB/PEG phase-change composite had stable dimensions and a snow-white appearance (Figure 3a, Table 1). Compared with DB, the weight of the DB/PEG phase-change composite increased, but the dimensions remain the same within experimental error. These results confirmed that DB is an ideal material for supporting organic PCMs. Owing to the excellent water solubility of PEG, the DB/PEG phase-change composite could withstand low-vacuum conditions without drying. The final composite held a stable shape after water evaporation at room temperature. As shown in Figure 3b, the XRD patterns of PEG, DB, and DB/PEG composite confirmed that the DB/PEG phase-change composite has an essentially even structure [26]. The FT-IR results (Figure 3d) are consistent with the XRD (Figure 3b); the spectra of DB/PEG composite included peaks at 2881 cm^−1^ (H–C–H, stretching vibration), 1468 cm^−1^ (H–C–H, plane bending vibration), 1113 cm^−1^ (C–O–C, stretching vibration), and 840 cm^−1^ (C–O–C, plane bending vibration) [27]. Compared with PEG, these peaks were expanded; therefore, the PEG were merely physically absorbed into the DB skeleton. This was confirmed by the thermal degradation results of TG and DTG (Figure 3e). As shown in Figure 3e, two peaks were observed for DB degradation, namely water evaporation (20 °C–140 °C) and degradation of cellulose (220 °C–380 °C). Only one peak was found in the ~300 °C–400 °C range for PEG, regardless of the molecular weight (800 or 1000) [11,28,29,30]. Mass loss of the DB/PEG composite was the same as that of DB and PEG; however, their peak temperatures and weight loss rates were different owing to physical interaction of DB and PEG. The main mass loss of the DB/PEG composite came from PEG, showing that DB was largely occupied by PEG. The simple physical interaction between PEG and DB without chemical reaction was beneficial to achieving reasonable temperature [31].

To confirm the loading amount of PEG, some DB samples were selected for analysis (Figure 3c). After delignification, nearly half of the mass was removed and the impregnation ration of PEG reached 81.9%; this indicates that PEG occupied a dominant position in the composite material and the role of function [32,33,34]. Compared with literature reports [35,36], the impregnation rate was much higher in this experiment. The main reason was that the cellulose matrix was rich in functional groups such as -OH, -NH, and -H; theoretically, hydrogen bonding could form between PEG and cellulose favoring PEG introgression (Figure 3d). It was for this reason that the experiment could be carried out under wet conditions without freeze-drying.

The structural characteristics of the DB sample were detected by employing Cryo-SEM. From comparison of Figure 4b,d, we can see that the cell wall of HTB was looser and more porous than that of original bamboo. Based on this, the HTB was much more easily soaked with chemical reagent, which was also the reason HTB was chosen as the matrix of DB [37,38,39]. The cell wall thicknesses of DB and HTB (Figure 4d,f) were measured by using image J and the values were 0.56 μm and 2.49 μm, respectively, confirming that the lignin and partial hemicellulose of HTB were removed during the multi-delignification process [40]. The cell wall of DB shrinkage rate was 77.5% by compared with that of HTB and gaps between the cells become larger, which provided channels for PEG impregnation. Figure 4g,h show cross-section images of the DB/PEG composite, all the pores of DB were completely filled with PEG polymer, forming a dense material [41]; meanwhile, the structure of the DB/PEG composite looked still like the original fiber state. In conclusion, the PEG was successfully introduced into the DB matrix, and a dense DB/PEG composite was created.

The phase transition temperature determines the application of PCM materials. In this work, DSC was used to evaluate the phase transition temperature of the DB/PEG phase-change composite (Table 2). From the curves shown in Figure 5a, the PEG and DB/PEG composite had obvious phase transition characteristics, including endothermic and exothermic peaks. The melting temperature (T*_m_*) of PEG-800 was 28.7 °C and the crystallization (T*_c_*) was 15.1 °C. The T*_m_* of PEG-1000 was 48.8 °C and the T*_c_* was 22.7 °C. After combination with DB, the T*_m_* (28.8 °C) of the DB/PEG composite was approximately the same as that of PEG-800, and two characteristic peaks were observed, but T*_c_* was 7.0 °C lower than that of PEG-800 or PEG-1000. Simultaneously, the ΔH*_m_* and ΔH*_c_* values were also lower than those of pure PEG. These results indicate that PEG was impregnated into DB to form a DB/PEG phase-change composite and that supercooling during the liquid–solid process resulted in the T*_c_* of DB/PEG decreasing to 7.0 °C [42]. However, the low temperature and high latent heat of the DB/PEG composite suggested that it was a good choice for developing indoor thermal energy storage materials. Compared with other reported materials, such as cellulose/PEG1000 (78.6 J/g; −5.4 °C–7.7 °C) [43], (cellulose nanocrystal) CNC/PEG2000 (82.3 J/g; 9.8 °C–47.1 °C) [43], and DW/PEG2000/CQD (carbon quantum dots; 137.6 J/g; 37.6 °C–49.3 °C) [10], the DB/PEG composite exhibited a suitable phase transition temperature (7.0 °C–41.6 °C) for human living spaces, ensuring a good latent heat of 84.3 J/g.

Duration of stable temperature is an important aspect in the practical application of energy storage materials. The change of phase-change temperature was detected as follows: selected samples were heated in the oven at 50 °C or cooled in the freezer at −20 °C; the tested temperature range was from 0 °C to 40 °C. From Figure 5b, the time taken for the temperature to increase from 0 °C to 40 °C was 25 min, and to decrease from 40 °C to 0 °C it was 25 min too. Compared with HTB, the heating time was 15 min, and the cooling time 20 min longer. This confirms that the DB/PEG composite has a good temperature maintenance effect. Meanwhile, the color changes of DB/PEG samples at different temperatures were recorded with an IR (Figure 5c), which confirmed that the temperatures remained stable.

The leakage was another important factor for PCM materials, to evaluate the stability of the DB/PEG composite, and its thermal resistance was detected under heating conditions from 25 °C to 60 °C. The obtained results are shown in Figure 5d; from the photograph of DB/PEG and PEG at different temperatures, it could be found that the DB/PEG composite showed excellent thermal stability during the temperature increasing process, and no liquid PEG polymer leakage. However, concerning the PEG polymer without the supported matrix, with the temperature increasing, its shape could not be kept constant. When the temperature reached 60 °C, the PEG polymer changed into liquid but the DB/PEG composite was still as in its original state. Hence, the DB/PEG composite has the potential to be used as a material for maintaining comfortable indoor environments without leakage; in turn, this could reduce the electrical energy consumption that is currently needed to maintain comfortable indoor temperatures.

## 4. Conclusions

In this study, a dimensionally stable DB/PEG phase-change composite was fabricated by impregnating PEG in a porous DB framework. Compared with HTB, 52.6% of the mass was removed by the extraction of lignin and hemicellulose to form the DB. Impregnation was carried out under wet conditions to ensure the regular arrangement of the DB structure and maintain its dimensional stability. A high impregnation rate of 81.9% was achieved owing to the physical interaction between the cellulose of DB and PEG such as hydrogen bonding. The enthalpy value was 72.2–84.3 J/g and the suitable phase-change temperature was ~7.0 °C–41.6 °C, confirming that the DB/PEG composite has the potential to be used as an energy storage material for indoor environments. Compared with the original HTB, the durations of rising and falling temperature of the DB/PEG composite were 15 and 10 min longer, respectively. This indicates that the DB/PEG composite is a promising material for adjusting temperature, especially for building comfortable ambient temperature, which would contribute to reducing human energy consumption.

## Figures and Tables

**Figure 1 polymers-15-01727-f001:**
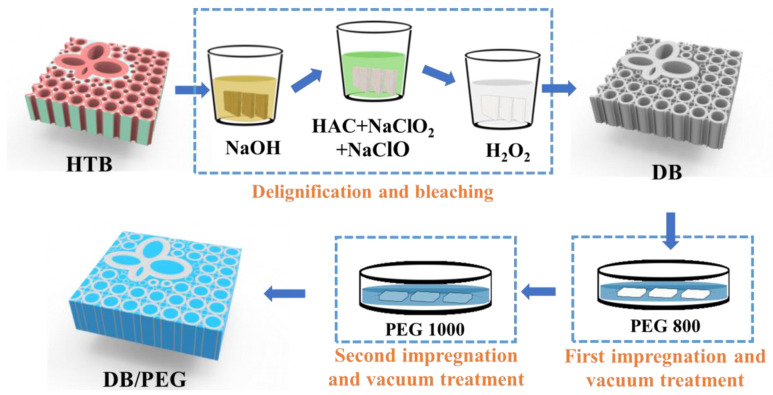
Preparation process of delignified bamboo (DB)/polyethylene glycol (PEG) composite. HTB, heat-treated bamboo.

**Figure 2 polymers-15-01727-f002:**
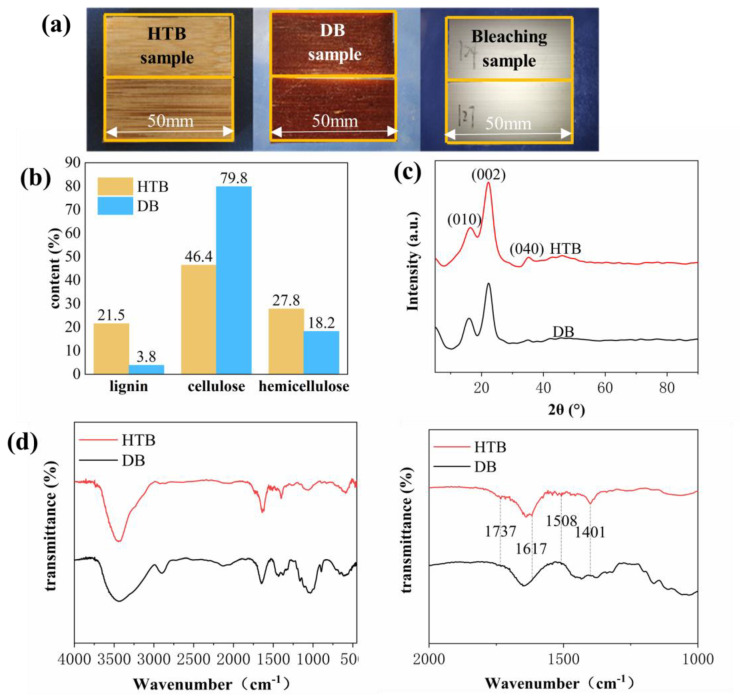
(**a**) Appearance of the heat-treated bamboo (HTB), delignified bamboo (DB), and bleached bamboo sample. (**b**) The content of lignin, cellulose, and hemicellulose in HTB and DB samples. (**c**) X-ray diffraction (XRD) results for HTB and DB samples. (**d**) FT-IR spectra of HTB and DB samples.

**Figure 3 polymers-15-01727-f003:**
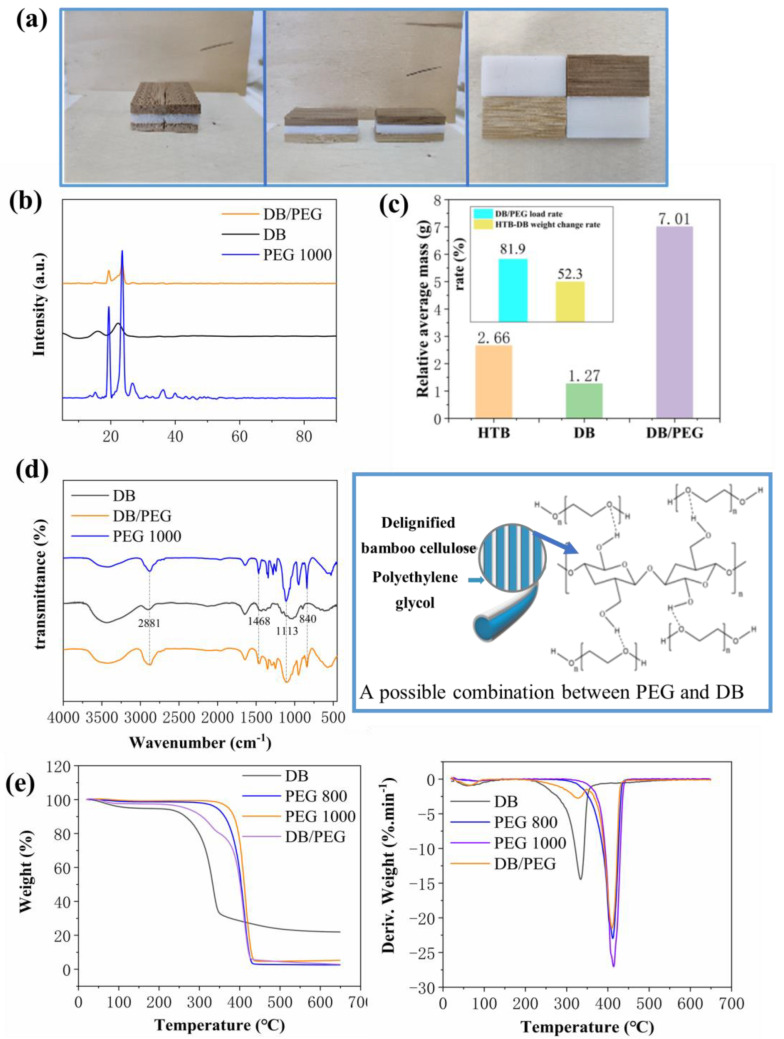
(**a**) Appearance of delignified bamboo (DB)/polyethylene glycol (PEG) composite from three directions. (**b**) X-ray diffraction (XRD) results for PEG, DB, and DB/PEG composite. (**c**) Loading amount of DB/PEG. (**d**) Thermogravimetry (TG) and derivative thermogravimetric (DTG) results for PEG, DB, and DB/PEG composite. (**e**) Measured Fourier transform infrared (FT-IR) spectra for PEG, DB, and DB/PEG composite and a schematic diagram for its possible combination.

**Figure 4 polymers-15-01727-f004:**
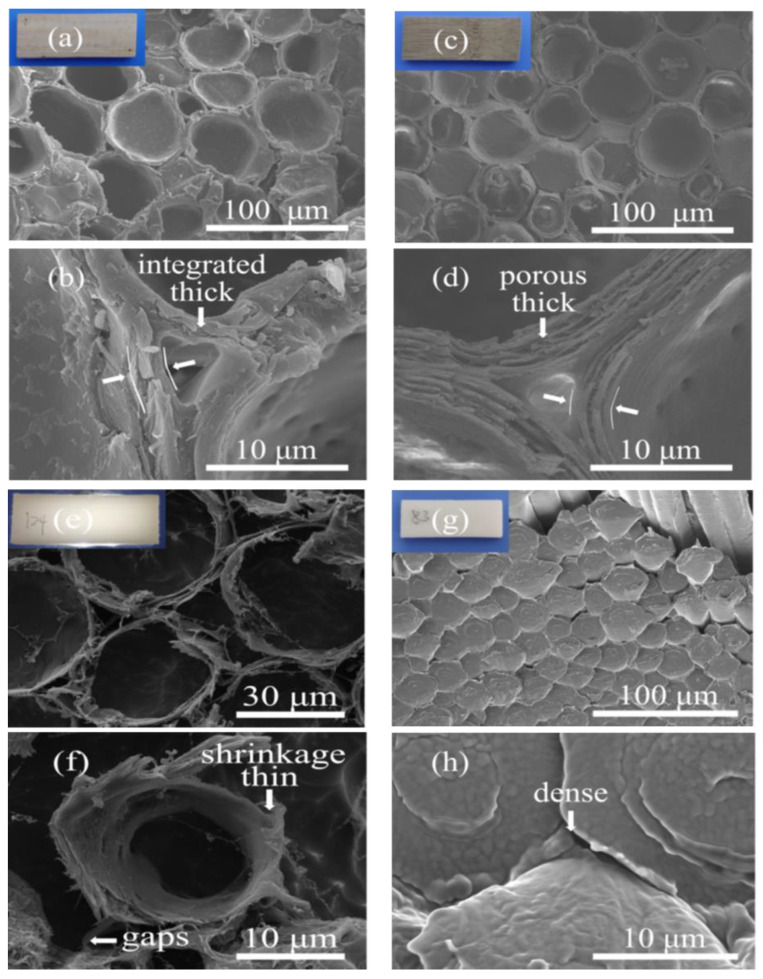
Scanning electron microscopy (SEM) image of samples. (**a**,**b**) cross-section image of original bamboo (OB) at different scales; (**c**,**d**) cross-section image of heat-treated bamboo (HTB) at different scales; (**e**,**f**) cryogenic SEM (Cryo-SEM) image of the delignified bamboo (DB) at different scales; (**g**,**h**) cross-section image of DB/PEG composite at different scales.

**Figure 5 polymers-15-01727-f005:**
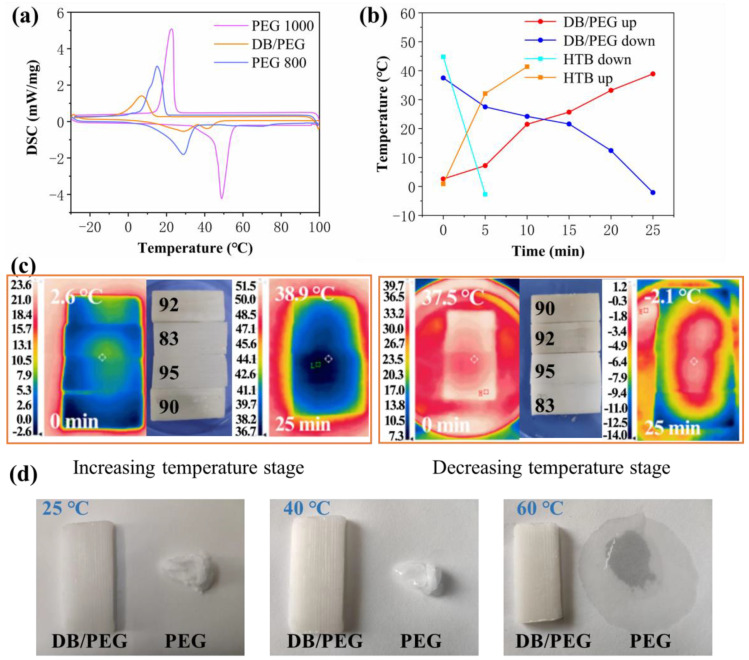
(**a**) Differential scanning calorimeter (DSC) curves of samples. (**b**) Duration of phase-change temperature in the range 0–40 °C via timer. (**c**) The IR pictures of tested samples at different temperature and their appearances. (**d**) The thermal stability of DB/PEG changing at room temperature to 60 °C.

**Table 1 polymers-15-01727-t001:** Measured dimensions and weights of samples.

Sample	Number	Weight (g)	Length (mm)	Width (mm)	Height (mm)	Weight Change Rate (%)	Length Change Rate (%)	Width Change Rate (%)	Height Change Rate (%)
HTB	95	3.96	49.65	21.32	5.26	44.5	0	7.5	0.4
DB/PEG		7.13	49.66	22.91	5.28				
HTB	83	2.46	49.92	21.24	5.28	64.1	0	1.3	0
DB/PEG		6.85	49.89	21.51	5.28				
HTB	90	2.72	49.98	21.31	5.39	61.0	0.2	1.9	0.9
DB/PEG		6.98	50.06	21.71	5.44				
HTB	92	2.43	49.90	21.44	5.33	66.3	0.4	2.8	5.4
DB/PEG		7.22	49.71	21.93	5.64				

**Table 2 polymers-15-01727-t002:** Phase-change temperature and enthalpy parameters.

Sample	T_m1_ (°C)	T_m2_ (°C)	ΔH_m_ (J/g)	T_c_ (°C)	ΔH_c_(J/g)
PEG-800	28.7	N/A	123.9	15.1	125.3
PEG-1000	48.8	N/A	157.7	22.7	148.1
DB/PEG	28.8	41.6	84.3	7.0	72.2

## Data Availability

Not applicable.

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
