# Peer review of "Dimensionally Stable Delignified Bamboo Matrix Phase-Change Composite under Ambient Temperature for Indoor Thermal Regulation"

_polymers, 2023, doi:10.3390/polym15071727_

Round 1
Reviewer 1 Report
Duan et al., developed a phase-change material based on delignified bamboo and polyethylene glycol to serve as an indoor thermal regulator. This research certainly is an interesting and widely investigated relevant topic. Therefore, this paper deserves recognition in this field of research. However, the authors should clearly motivate certain choices made and also go critically through statements in the text. There are also some grammar and spelling mistakes to be found throughout the text (as well as missing spaces and such). The authors should go carefully through the text to correct these errors. Overall, the manuscript merits publication in Polymers but there are some major and minor comments which ought to be addressed before publication:
- - What to do the authors mean with “not enough friendly” in the abstract section in following sentence: “. For application, the PCM is often supported with a synthetic porous material which is not enough friendly for people and housing environment”. Please reformulate.
- - Please reformulate the first sentence of the introduction section. It seems that the word “with” is not on its place there.
- - The authors use keywords as “renewable”, “environmentally friendly” when talking about bamboo, what is indeed applicable. However, on the other hand, the authors apply a chlorine based delignification technique that is not environmentally friendly (even highly toxic). Why did the authors opt for this delignification method instead of other (more environmentally friendly) delignification techniques? This should be motivated in the text.
- - In section 2.4 characterization techniques, it should be sulfuric acid instead of sulfide acid.
- - As the composition of bamboo (besides lignin) is highly variable based on several parameters (age, geographic location, species, etc) it is highly advisable to perform a more thorough characterization (also determining cellulose and hemicellulose).
- - Can the authors correctly state this (page 6)? “Confirming that the lignin and hemicellulose of HTB were removed during the multi-delignification process”. Normally chlorine based delignification methods do not remove hemicellulose in its entirety. Did the authors perform a chemical analysis to confirm this?
- - Please improve the quality of the caption of Figure 5.
Author Response
Dear reviewer:
Thank you for your comments on this manuscript, I am very grateful to you.
I have carefully revised the manuscript according to the advice and detail comments. And the revised parts has been marked with blue colour in revised manuscript. I really hope that the level of this revised manuscript have been substantially improved and is suitable for publication, check them again please. If anything is not clear, please not hesitate to tell us to modify further. Thank you very much for giving us this opportunity.
According to the comments, we have carefully revised the manuscript, and the main corrections and the responds are as following:
All this question were responds in respone lettter rewivewer 1.docx
Thank you!

Reviewer 2 Report
This is an interesting study and the paper is well-written. Some comments:
Introduction
“A variety of delignified methods have been reported; however, only a limited amount of bamboo can be treated, making it difficult to accommodate wide-spread applications for bamboo[18-20]” Can you elaborate what is the problem encountered in treating bamboo?
The hypothesis of the study is not study. The objectives of the study are not clear, why heated treated bamboo bricks, what do you expect from this? Please clarify
Section 2.3
200 MPa – just to confirm, is this the vacuum pressure? Isn’t it high?
What is the apparatus you used for vacuum treatment?
Section 2.4
Please separate each testing into subsection
Please revise the font in Figure 5.
The study focuses on the prevention of leakage, do the authors perform any test to prove that DB is a porous structure that can overcome the leakage of PEG?
Author Response
Dear reviewer:
Thank you for your comments on this manuscript, I am very grateful to you.
I have carefully revised the manuscript according to the advice and detail comments. And the revised parts has been marked with blue colour in revised manuscript. I really hope that the level of this revised manuscript have been substantially improved and is suitable for publication, check them again please. If anything is not clear, please not hesitate to tell us to modify further. Thank you very much for giving us this opportunity.
According to the comments, we have carefully revised the manuscript, and the main corrections and the responds are as following:
All this question were responds in respone lettter rewivewer 2.docx
Thank you!

Round 2
Reviewer 1 Report
The authors addressed most of the comments and questions raised by the reviewer. However, the reviewer has 2 major comments/remarks left:
1/ Please go carefully through the text and check for English grammar and spelling mistakes. It might be usefull to proofread this article by a native English speaker.
2/ The authors used a method that for the determination of cellullose and hemicellulose. Is this an in-house method? Or is this derived from the literature. If the first, please indicate specifically in the text. If the latter, please provide the readers/reviewer with a reference.
Author Response
Dear reviewer:
Thank you for your comments on this manuscript, I am very grateful to you.
I have carefully revised the manuscript according to the advice and detail comments. And the revised parts has been marked with blue colour in revised manuscript. I really hope that the level of this revised manuscript have been substantially improved and is suitable for publication, check them again please. If anything is not clear, please not hesitate to tell us to modify further. Thank you very much for giving us this opportunity.
According to the comments, we have carefully revised the manuscript, and the main corrections and the responds are as following:
All this question were responds in respone lettter rewivewer 1-2.docx
Thank you!

Round 3
Reviewer 1 Report
The authors addressed the reviewers comments and therefore the reviewer accepts this manuscript for publication in Polymers.